# Longitudinal Study of Bone Height Change between Two Approaches for Sinus Floor Elevation

**DOI:** 10.3390/medicina59061132

**Published:** 2023-06-12

**Authors:** Jun-Hyeong Park, Yong-Gun Kim, Jo-Young Suh, Du-Hyeong Lee, Jin-Wook Kim, Jae-Mok Lee

**Affiliations:** 1Department of Periodontology, Kyungpook National University School of Dentistry, Daegu 41940, Republic of Korea; 2Department of Prosthodontics, Kyungpook National University School of Dentistry, Daegu 41940, Republic of Korea; deweylee@knu.ac.kr; 3Department of Oral and Maxillofacial Surgery, Kyungpook National University School of Dentistry, Daegu 41940, Republic of Korea

**Keywords:** dental implant, bone loss, maxillary sinus elevation

## Abstract

*Background and Objectives*: The purpose of this study is to assess the long-term maintenance of each approach of sinus elevation, the crestal approach and lateral approach, by comparing the radiographic results of each technique. *Materials and Methods*: In total, 103 patients who had undergone an implant procedure with either the crestal approach or lateral approach method applied to their maxillary molar edentulous area were included. Using orthopantomographs, the radiographic changes were consistently evaluated over 3 years after the procedure (immediately after procedure and 1 year, 2 years and 3 years after implant placement) *Results*: The radiographic evaluation after 3 years of implantation with sinus elevation showed a significant amount of bone formation (8.07 mm for crestal approach and 12.00 mm for lateral approach method). The largest amount of grafted height loss occurred during the 1 year, but the resorption was minimal (0.98 mm for crestal approach and 0.95 mm for lateral approach method) over the entire 3 years. *Conclusions*: Although the lateral approach showed more bone growth, the amount of bone resorption was similar to that of the crestal approach. Both methods showed the highest amount of bone resorption in the first year, and the amount of change thereafter was insignificant. It is judged that both methods can be used according to the situation to help implant placement.

## 1. Introduction

Implant placement in maxillary molars requires a variety of considerations and presents several clinical challenges [1]. The most obvious difficulties are the relationships with the maxillary sinus, the resulting lack of bone height and bone resorption after a traumatic procedure or due to inflammatory causes [2,3,4]. Successful implant placement cannot be expected without adequate bone height.

Maxillary sinus floor elevation was initially described by Tatum at the Alabama implant conference in 1976, with a paper on the concept subsequently published by Boyne and James in 1980 [5]. Its development stemmed from the necessity to restore the posterior maxilla using implants. This procedure is one of the most common pre-prosthetic surgeries performed in dentistry today. Since its first description, numerous articles on different grafting materials, modifications to the classic technique, and comparisons among different techniques have been published [6,7,8].

Two main approaches for performing the maxillary sinus floor elevation procedure have been described in the literature. The first is the lateral approach procedure that was originally described by Tatum. It is a classic approach and is the most commonly performed technique [9,10]. More recently, Summers presented a second approach [8], called the crestal approach, using osteotomes. The crestal approach is considered to be a more conservative method for sinus floor elevation.

The lateral approach procedure begins with a crestal incision made on the alveolar ridge. Once the flap has been raised to the desired level, a U-shaped trapdoor is created on the lateral buttress of the maxilla with a round burr. The height of this trapdoor should not exceed the width of the sinus to allow for the final horizontal position of the new sinus floor. The sinus membrane is then gently lifted from the bony floor using an antral curette. It is important to release the sinus membrane in all directions, namely the anterior, posterior, and medial directions, before attempting to intrude on the trapdoor medially. A space is created after the elevation of the sinus membrane by the intruded trapdoor. This space is then grafted with different materials to provide the platform for implant placement. Care should be taken not to overfill the recipient site, as this will cause membrane necrosis. Implants are placed either simultaneously with the graft (one-stage lateral approach procedure) or after a period of up to 12 months to allow for graft maturation (two-stage lateral approach procedure). The initial bone thickness at the alveolar ridge seems to be a reliable indicator to help decide between these two methods. If the bone thickness is 4 mm or less, initial implant stability would be jeopardized. Hence, a two-stage lateral approach procedure should be performed, but the reverse is true for a one-stage procedure. The one-stage procedure is less time-consuming for both the clinician and the patient. However, it is more technique-sensitive and its success relies heavily on the amount of residual bone.

One of the drawbacks of the lateral approach procedure is that it requires the raising of a large flap for surgical access. Summers proposed a conservative crestal approach using osteotomes for maxillary sinus floor elevation in 1994 [8]. The goal with the approach is to extend the instruments just shy of the sinus membrane. Osteotomes of increasing sizes are introduced sequentially to expand the alveolus. With each insertion of a larger osteotome, the bone is compressed, and pushed laterally and apically. This is performed until the osteotome can be inserted close to the sinus floor. Once the largest osteotome has been inserted into the implant site, a prepared bone mix is filled into the osteotomy site as the grafting material. The final stage of sinus floor elevation is completed by reinserting the largest osteotome into the implant site with the graft material in place. This causes the added bone mix to exert pressure on the sinus membrane and elevate it. Additional grafting material can subsequently be added and tapped in to achieve the desired amount of elevation. Once the desired height is attained, the implant fixture is inserted. The implant fixture should be slightly larger in diameter than the osteotomy site created by the largest osteotome. It becomes the final osteotome, “tenting” the elevated maxillary sinus membrane. 

From various studies, lateral approach sinus elevation is considered to create more bone height than the crestal approach, especially for two-stage lateral approach sinus elevation. However, data for evaluating the long-term comparison between the two procedures so far is scarce. In this study, we tracked cases of crestal and the lateral sinus elevation for a term of up to 3 years to evaluate their effectiveness through radiographic comparison.

## 2. Materials and Methods

### 2.1. Patient Selection

From 250 patients who visited the Department of Periodontics of Kyungpook National University Hospital for implant installation and sinus floor elevation, 220 sites were selected. Patients with five kinds of OPTs were selected, which were OPT before surgery, OPT immediately after surgery, OPT after 1 year, OPT after 2 years, and OPT after 3 years. Of the 220 sites screened, sites in 103 patients qualified for the analysis. The most common reason for exclusion was the lack of panoramic radiographs during the follow-up periods and the lack of visible bone material in the radiographic images.

The patients provided informed consent to participate in this clinical study. In the previous 12 months, no symptoms of sinus distress or systemic disease were seen in any of the patients. Forty patients whose native bone height was above 4 mm underwent the crestal approach procedure with the simultaneous placement of an implant, whereas other patients who had a native bone height less than 4 mm underwent lateral window sinus elevation with the delayed placement of an implant. The study protocol was reviewed and accepted by the Research Ethics Committee at Kyungpook National University (KNUDH-2022-12-02-00).

### 2.2. Operative Technique

#### 2.2.1. The Crestal Approach

Full-thickness flaps were elevated following a crestal incision under local anesthesia. If necessary, vertical releasing incisions were made to elevate the flap to the mucogingival junction. The initial osteotomy was performed using a high-speed carbide round burr, a 2-mm twist drill, or manually using an osteotome (XiVE; Friadent, Mannheim, Germany, 3I; Palm Beach Gardens, FL, USA). This was followed by different-sized osteotomes to expand the alveolar bone. Various prepared bone materials that act as shock absorbers were added to the preparation site with a carrier. At this stage, the Valsalva maneuver, performed by forcibly exhaling against closed lips and a pinched nose, was used to confirm the integrity of the sinus membrane.

#### 2.2.2. The Lateral Approach

A full-thickness flap was raised, and the lateral wall of the maxillary sinus was exposed. An oval-shaped lateral window was created using a round diamond burr. The sinus membrane was reflected, and a space was created. If the membrane was perforated or torn, a collagen membrane was used to repair the damage. If no tears were found, the graft material was placed. The area was then sutured.

### 2.3. Bone Augmentation Materials

Three different graft types were used, namely xenograft, alloplast, and autograft. Anorganic bovine bones (BioOss; Geistlich AG, Wolhusen, Switzerland and OCSB; NIBEC, Seoul, Republic of Korea) were widely used as xenografts. Bioactive glass (Biogran; Implant innovations Inc., Palm Beach Gardens, FL, USA) and bioceramics (MBCP; Biomatalante Sarl, Nantes, France) were used as alloplastic materials. Oragraft (LifeNet; Virginia Beach, VA, USA) was frequently used as an autograft. Most of the autogenous bone was taken from the trephine core while drilling. However, alloplastic and autograft materials were only used in 3 cases and in 1 case, respectively, so that only the xenograft type (BioOss; Geistlich AG, Wolhusen, Switzerland and OCSB; NIBEC, Seoul, Republic of Korea) was considered.

### 2.4. Postoperative Care

Postoperatively, the patients were instructed to rinse their mouths twice a day with a 0.12% chlorhexidine solution for 10 days. Patients were asked to continue taking amoxicillin (500 mg) with clavulanate potassium (125 mg) 3 times a day for 5 to 7 days, and the sutures were removed after 10 to 12 days. For patients of the lateral approach group with sinus grafting alone, implants were placed 6 to 8 months postoperatively, depending on the implant fixture and radiographic analysis. If implants had been placed at the time of the sinus augmentation, they were allowed to osseointegrated for 6 to 8 months before they were uncovered.

### 2.5. Radiographic Analysis

For quantitative analysis of panoramic radiographs, Orthophos CD (SIEMENS, Germany) program No. 11 was used. The height of the graft material was measured and calculated, with a mean of 25% conversion factor adjusted for the magnification error. The magnification of the panoramic radiograph was also corrected using the known actual length of the inserted implants.

The amount of native bone was quantified and stratified into three groups based on the amount of bone present: <5 mm, between 5 and 8 mm, and >8 mm for the crestal approach group; and <2 mm, between 2 and 4 mm, and >4 mm for the lateral approach group.

Using a mouse, a cursor was placed on top of each implant. A similar measurement could be taken from the height of the graft material using the top of the implant as a standard reference point. To evaluate the change in graft height, these analyses were repeated during the follow-up period. This was performed by one investigator not involved in the treatment of patients. For each patient, four panoramic radiographs were taken into consideration, one panoramic radiograph was taken directly after implant placement, and the others were taken at 1 year, 2 years and 3 years after implant placement.

### 2.6. Statistical Analysis

All data analyses were performed using a statistical software package (SPSS 17.0 KO; SPSS Inc., Chicago, IL, USA). Student’s t test was used to compare the crestal approach and the lateral approach groups. The Friedman test was used to analyze the changes in the grafted bone height at the baseline, 1 year, 2 years and 3 years after implant insertion in each group. The Wilcoxon signed-rank test was used to establish the time points when significant changes occurred. The Scheffe post hoc test was also used. *p* < 0.05 was considered statistically significant.

## 3. Results

### 3.1. Crestal and Lateral Approach Sinus Augmentation

The mean gain in graft height using the crestal approach was 8.07 mm and that of the lateral approach was 12.00 mm. A statistically significant difference between the presurgical and postsurgical average bone height existed in only the lateral approach techniques (*p* = 0.002). The reduction in height after a 1-year follow-up was 0.58 mm for the crestal approach group and 0.62 mm for the lateral approach group. Moreover, the differences between the two groups were not statistically significant (*p* = 0.869). The reduction in height after a 3-year follow-up was 0.98 mm for the crestal approach group and 0.95 mm for the lateral approach group, which were also not statistically significant (*p* = 0.921) (Table 1; Figure 1).

### 3.2. Native Bone Height

The amount of native bone did not have a significant effect on the change in graft height (crestal approach group: *p* = 0.90; lateral approach group: *p* = 0.88). The least amount of graft loss was found in the crestal approach group with a native bone height below 5 mm (0.65 mm). It was apparent that the mean gain in alveolar height was inversely proportional to the initial bone height in both methods (Figure 2 and Figure 3; Table 2 and Table 3).

### 3.3. Follow-Up Period

In both groups, the largest amount of reduction occurred during the first year (crestal approach: 77.2%; lateral approach: 66.6%). There was a statistically significant reduction in the grafted bone height between all the time points (*p* = 0.00). The mean change in the grafted bone height tended to decrease with time. In the crestal approach group with between 5- and 8-mm bone height and the lateral approach group with bone height greater than 2 mm, the differences in graft height between all the time points were statistically significant (*p* < 0.05). In the crestal approach group with greater than 8-mm bone height, changes from baseline to 1 year and from 1 to 2 years revealed statistically significant differences (*p* = 0.001 and 0.005, respectively) (Figure 2 and Figure 3; Table 2 and Table 3).

## 4. Discussion

Adequate bone height with or without graft materials is crucial for implant maintenance. Therefore, bone gain through available methods is important. In a previous study, the bone height gain was larger in the lateral approach method [11]. This is because the lateral approach allows for a larger surgical field than the crestal approach, along with a wider surgical field of view, making it a common choice when the existing bone height is low. In general, if there is no need for a large increase in bone height, the crestal approach is more non-invasive, and if a large amount of bone grafting is required, the lateral approach is the gold standard. In this study, the lateral approach was chosen in all cases with a very low bone height of 4 mm or less, and the crestal approach was chosen in most cases above 4 mm. In some cases, the lateral approach was chosen for anatomical structures and tooth arrangements that are difficult to access with the crestal approach.

For long-term implant maintenance, changes in bone resorption after bone augmentation are also important. In this study, there was no significant difference between the two methods. These results indicate that both methods can be useful for sinus elevation and are options that can be accessed without hesitation, which is consistently stated by other authors [12,13,14,15].

In this study, a radiographical and technical error was introduced by the operator in measuring the bone height. However, only one trained evaluator participated. The radiographical error was sufficiently corrected using the actual implant length ratio. Additionally, this approach is considered the best as it is the only available method for measuring bone height. Although computed tomography (CT) has become widely commercialized these days and is the basis for most implant procedures, this trend was not common among the practitioners of the research record used. Therefore, a comparison with a panoramic radiograph is the most reliable comparison. In additional studies in the future, active use of CT is recommended for more accurate and visible comparison.

Before performing the procedure, analyzing the bone height and surrounding shape with radiographs is necessary as marginal bone loss can be the most important determining factor in implant dentistry [16,17,18]. In the consensus conference of 1996 on sinus grafts, it was recommended that sinus floor elevation should be performed if the residual bone measures 8 mm or less [19]. In contrast, Rosen et al. [20] stated that procedures on residual bone of at least 4 mm can be highly successful. Another study revealed that the most effective bone height for the crestal approach was between 4 and 8 mm, while that for the lateral approach was between 2 and 4 mm [10]. The amount of change according to the bone height in the crestal approach was not significant. In the lateral approach, only the amount of change in the native bone height of between 2–4 mm was significant. Regardless of the significance, bone with a height of between 5–8 mm in the crestal approach and 2–4 mm in the lateral approach showed the largest change in bone height.

Although the difference in bone height increase according to the type of bone graft materials should be considered, most of the selected patients routinely used only one material (BioOss; Geistlich AG, Wolhusen, Switzerland and OCSB; NIBEC, Seoul, Republic of Korea), and the number of patients using other materials was very small, three patients and one patient, respectively, so a meaningful comparison was not made. It seems necessary to consider the change in bone height for each type of bone graft material in additional research.

Additionally, an average decrease in bone height of 0.98 mm in the crestal approach and 0.95 mm in the lateral approach was noted, which is seen as a part of the adaptation process after the prosthesis is installed [21]. Most of the bone loss occurred within the first year after implant placement (after prosthesis installation). This is believed to be due to bone loss caused by various factors during the first year, such as surgical trauma, occlusal overload, microgap, biological width, and crest module [22]. The decrease in the amount of change in bone height after the initial bone loss can be attributed to an increase in bone density and a decrease in loading [23]. Although it is a radiologic comparison, this study shows that both the crestal and lateral approaches have good bone graft prognosis within 1 mm. There will be differences based on radiographic error and the actual degree of osseointegration, but even after accounting for these, these numbers are unlikely to have a significant impact on the prognosis of sinus elevation. Therefore, it is likely that the type of implant screw or operator’s skills and bone quality are more important factors in the prognosis of implants with sinus elevation.

In terms of implant success rates, there is no significant difference between the crestal and lateral approaches. In a study comparing ISQ values, it was found that both methods showed good values of between 60–65 [24,25,26], with no significant difference between the methods [27]. In addition, ISQ values increased over time, which is an indicator of successful implant placement [28]. Furthermore, when examining the long-term success rate of implants with sinus elevation, a success rate of approximately 98.3% was observed [29]. In the case of failed implants, failure is more likely to occur within 5 years, and is characteristically seen within 1 year and between 3–4 years. Although ISQ values were not available in the records of this study, the very low implant failure rate suggests that there would be no significant difference in ISQ values and implant failure rates between the two methods, and that bone height reduction in both methods may not play a significant role in implant prognosis in this study.

If there is no change in the difference in bone loss between the two methods, it is entirely up to the operator which technique to choose. Unlike the lateral approach, the crestal approach is less invasive and can expect a quick healing period. In addition, since the procedure is performed while securing a relatively high residual bone height, there is an advantage in that the burden on the operator can be slightly reduced. On the other hand, the surgeon should be aware that a large amount of bone graft cannot be performed because a narrow area is used when approaching with the crestal approach. The lateral approach is a procedure that can be a little more sensitive to changes in bone height because it can perform a large amount of bone grafting with a relatively small bone height. However, since the difference from the crestal approach is insignificant, it is thought that it can be approached without much burden depending on the skill level of the operator. It will be helpful for the surgeon to recognize that both techniques can be flexibly selected according to the remaining bone height to secure the bone height for implant placement, and that there is no significant difference in the prognosis. This is in line with the results of another article [30]. However, it is important to note that there are various complications that can arise from procedures. The most common complication is sinus membrane perforation, which occurs in 5.13% of cases in the crestal approach and 17.75% in the lateral approach [31] Other complications include graft exposure, sinusitis, and various clinical symptoms. As shown in this article, complications are less likely to occur in the crestal approach than in the lateral approach, and the severity and types of complications are less. When a sinus membrane perforation occurs, resorbable collagen membranes or collagen surgical dressings are the most commonly known methods, but more recently, platelet-rich fibrin (PRF) clots [32] have been increasingly used. In addition, other complications (maxillary sinusitis, wound dehiscence, infection, etc.) are treated with antibiotics and chlorhexidine irrigations. Therefore, it seems that it is more important for practitioners to consider the risks of procedures and the possibility of complications rather than the amount of change in bone height, and it is imperative that they are fully aware of how to deal with them.

## 5. Conclusions

The lateral approach showed more bone growth, but the amount of bone resorption was similar to that with the crestal approach. Both methods showed the highest amount of bone resorption in the first year. However, the amount of change thereafter was insignificant. It is judged that both methods can be used depending on the situation to assist in the placement of the implant.

## Figures and Tables

**Figure 1 medicina-59-01132-f001:**
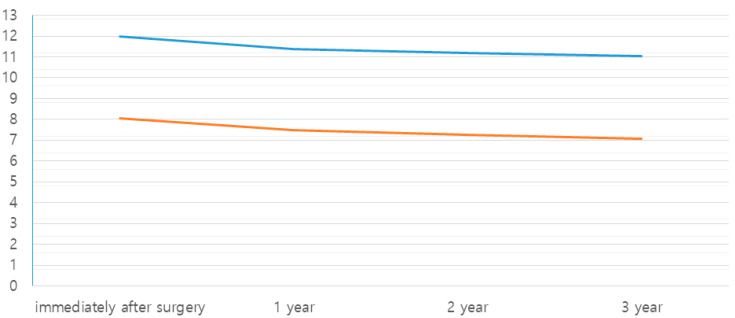
Comparison of crestal and lateral approach groups by follow-up period (mm).

**Figure 2 medicina-59-01132-f002:**
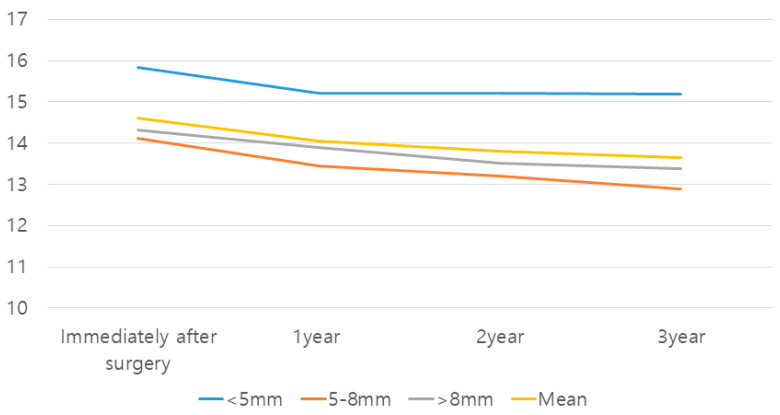
Estimated mean change in the grafted bone height in the crestal approach group.

**Figure 3 medicina-59-01132-f003:**
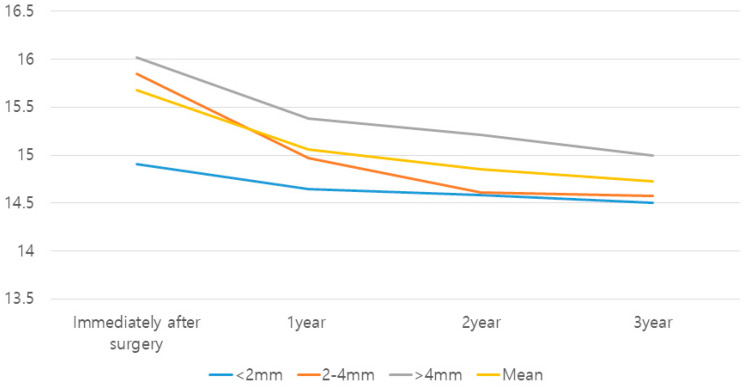
Estimated mean change in the grafted bone height in the lateral approach group.

**Table 1 medicina-59-01132-t001:** Estimated mean change in the grafted bone height in both groups.

	Mean Native Bone Height (mm)	No. of Implants	Gain in Bone Height (mm)	Change in the Grafted Bone Height (mm)		
				1 year	2 year	3 year
Crestal	6.55	40	8.07	−0.58	−0.80 (−0.22)	−0.98 (−0.18)
Lateral	3.68	63	12.00	−0.62	−0.83 (−0.21)	−0.95 (−0.12)
Total	5.12	103	10.47	−0.60	−0.82 (−0.22)	−0.96 (−0.14)

Values are presented as numbers (mm).

**Table 2 medicina-59-01132-t002:** Radiographic bone height changes in the crestal approach group.

Native Bone Height (mm)	Mean Height (mm)	No. of Implants	Gain in Bone Height (mm)	Change in the Grafted Bone Height (mm)		
				1 Year	2 Year	3 Year
<5 mm	3.48	10	12.35	−0.62	−0.61	−0.65
5–8 mm	6.11	16	8.01	−0.68	−0.93	−1.23
>8 mm	9.25	14	5.07	−0.43	−0.81	−0.93
Mean	6.55	40	8.07	−0.58	−0.81	−0.98

Values are presented as numbers (mm).

**Table 3 medicina-59-01132-t003:** Radiographic bone height changes in the lateral approach group.

Native Bone Height (mm)	Mean Height (mm)	No. of Implants	Gain in Bone Height (mm)	Change in the Grafted Bone Height (mm)		
				1 Year	2 Year	3 Year
<2 mm	1.34	16	13.57	−0.26	−0.32	−0.40
2–4 mm	3.15	21	12.69	−0.87	−1.23	−1.27
>4 mm	5.55	26	10.47	−0.63	−0.81	−1.02
Mean	3.68	63	12.00	−0.62	−0.83	−0.95

Values are presented as numbers (mm).

## Data Availability

The datasets used and/or analyzed during the current study are available from the corresponding author upon reasonable request.

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
