# Peer review of "Longitudinal Study of Bone Height Change between Two Approaches for Sinus Floor Elevation"

_medicina, 2023, doi:10.3390/medicina59061132_

Round 1
Reviewer 1 Report
Dear Authors
In the presented work, it is proposed to evaluate a three-dimensional structure using a two-dimensional image. The result showed no significant ones.
He does not negate the use of panoramic photos, because for many years they have been and still are a valuable tool. But I don't think so in this type of research work.
I think that the idea for the work is good, but the methodology needs to be refined
Sincerely Yours
Reviewer
Author Response
Thank you for your comments. I fully recognized that this study used two-dimensional data but it is widely used method for evaluating bone height. In future studies, a three-dimensional approach will be needed and available.

Reviewer 2 Report
Dear Authors,
Interesting topic of the article.
Abstract: Well written.
The keywords are to be arranged in alphabetical order.
Introduction:
- The paragraph between lines 42-64 requires references.
- Row 80'' various studies''.... references
Aim: to rewrite it in a more concise, more convincing manner...as you wrote it in the abstract
Material and method
- Wich are the inclusion/selection criteria?
- The text between lines 88-98 must be rearranged.
- Who carried out the interventions? How many operators?
- Row 114 – name/type of burr?
- Rand 116 – what kind of membrane (company)?
- Bone augmentation... do you have a record of the material applied to each patient?
- What kind of implants were applied (company)?
- How many evaluators participated in the radiological determinations?
- Do you have intraoperative images?
Results: well written, it may be worth leaving only the tables.
Discussion: What are the limitations of the study?
Reviewer 3 Report
This is an interesting study. However, the presentation can be improved.
1. The last two sentences in the abstract should be re written as the sentence is not scientifically sound
2. The limitation of the study should be detailed in more
3. What are the implications and future considerations if a similar study is conducted.- this could be added in discussions
4. Will the integration of omics or AI will lead to a better understanding of crestal bone loss or other factors affecting bone loss? This can also be included in the discussion
5. Please provide ethical approval no and date in the manuscript method section
5. A figure showing how the bone loss was measured as example would be helpful for readers
6. The writing requires a bit of english modification.
Round 2
Reviewer 1 Report
Dear Authors
I accept Your answer
The paper may be published
Sincerely yours
Reviewer
Author Response
Thank you for your review.